# Compliance with COVID-19 Prevention Measures in the Spanish Population during the New Normal: Will the Need for Greater Community Involvement Be One of the Lessons Learned?

**DOI:** 10.3390/ijerph192315983

**Published:** 2022-11-30

**Authors:** Ana María Recio-Vivas, José Miguel Mansilla-Domíngez, Ángel Belzunegui-Eraso, David Peña-Otero, David Díaz-Pérez, Laura Lorenzo-Allegue, Isabel Font-Jiménez

**Affiliations:** 1Department of Nursing, Faculty of Biomedical and Health Science, Universidad Europea de Madrid, 28670 Madrid, Spain; 2Medical Anthropology Research Center, Faculty of Nursing, Rovira i Virgili University, 43002 Tarragona, Spain; 3Sub-Directorate of Caring, Cantabrian Health Service, Material Resources Unit, Management of Products, Equipment and Health Technology, Hospital Universitario Marqués de Valdecilla, 39011 Cantabria, Spain; 4Nursing Area, IDIVAL Research Institutes Valdecilla, 39011 Cantabria, Spain; 5Respiratory Nursing Department at Sociedad Española de Neumología y Cirugía Torácica (SEPAR), 08029 Barcelona, Spain; 6Pneumology and Thoracic Surgery Service of the Hospital Universitario Nuestra Señora de Candelaria, 38010 Santa Cruz de Tenerife, Spain

**Keywords:** COVID-19, SARS-CoV-2, health behavior, public health, protective measures

## Abstract

Throughout the pandemic, national and international health authorities have called on the population to collaborate and contribute with their behavior to control the problem. The aim of this study is to analyze the implementation of the protective measures against COVID-19 and to determine the factors involved in their compliance. To respond to the objectives, a cross-sectional study was performed involving a total of 5560 individuals. An ad hoc online questionnaire was created and shared through social networks, scientific societies, and various health institutions. The probability of high or total compliance with the protective measures was higher in women (OR = 1.401) and as age increases, with an OR = 2.524 in the interval between 31 and 64 years old and an OR = 2.896 in the oldest interval (65 and over). This study shows the characteristics of the population that considers it more likely to be infected by SARS-CoV-2, thus adopting greater adherence to prevention measures. Knowing which factors are associated with adherence to protective measures is essential for establishing effective pandemic control measures. Our findings may be useful for designing future awareness campaigns adapted to different socio-demographic characteristics in settings affected by COVID-19.

## 1. Introduction

In December 2019, in Wuhan (China), a series of severe acute respiratory syndrome cases were described, caused by a previously unknown etiological agent, a virus of the Coronaviridae family, which was named SARS-CoV-2 (Severe Acute Respiratory Syndrome Coronavirus 2) [1]. From there, the virus spread to other countries, until on 11 March 2020 the WHO declared COVID-19, a disease caused by the aforementioned virus, a pandemic. The first case of COVID-19 in Spain was reported on 31 January 2020. Since then, the number of patients infected by SARS-CoV-2 increased exponentially, until a state of alarm was declared in several countries on 14 March 2020.

In Spain, the COVID-19 outbreak hit hard, leading to a strict containment of the population from 14 March to 21 June 2020. This confinement was accompanied by a series of measures to contain and prevent the spread of SARS-CoV-2, which changed over time and varied between the different autonomous communities that make up the country, according to the particular epidemiological situation in each of them [2].

According to the study published in November 2020 by the National Epidemiology Centre (CNE), which is based on data obtained from the National Epidemiological Surveillance Network (RENAVE) and the Daily Mortality Monitoring System (MoMo), the lethality of SARS-CoV-2 infection in the non-institutionalized Spanish population during the first months of the pandemic (i.e., December 2019 to March 2020), was between 0.3% and 1% [3].

These data show the challenge faced by authorities and health professionals during the first wave of the disease, and how the development of the disease provided new data [4]. This study focuses on the period after confinement, which the media called the “new normal”.

In the absence of definitive treatment and without a fully implemented vaccination programme, the fight against COVID-19 at the time of our study was based on behavioral changes brought about by preventive measures.

Knowledge about disease and positive attitudes towards preventive measures play a key role in controlling the spread of infection diseases such as SARS-CoV-2 [5], increasing the willingness to adopt such measures and to collaborate in them [6]. 

These measures will only be successful with the cooperation of the general population [7] who must also comply with other protective measures, such as the use of a mask in public spaces, where the risk of infection is higher, and proper hygiene. Besides, proper health education is essential to identify the symptoms indicative of the COVID-19 infection and act appropriately upon them. 

Likewise, the level of perceived risk associated with the disease is associated with adherence to preventive measures, and in a directly proportional manner, people who perceive greater risk are more likely to adopt protective measures [8]. 

In this scenario, motivation and understanding are vital for proper compliance with protective measures by the general population [9]. It is therefore necessary to identify the factors that act as a barrier to effective implementation of protective measures, such as the use of masks, hand hygiene and physical distancing. It is known that socio-demographic and psychological factors can influence how the recommended protective measures are perceived. Therefore, prevention strategies designed to stop the spread of the disease must include these factors. 

Throughout the pandemic, and in particular during the period in which this article is being written, national and international health authorities have called on the population to collaborate and contribute with their behavior in order to control the problem [10]. 

The emphasis on individual responsibility has relegated community participation, which is based on the need to involve communities in decision-making, to the background [11]. Many community actions have been carried out to boost the cooperation and solidarity of their members. Even so, despite clear evidence of their importance in terms of behavior, equity, and health promotion in health crisis situations [12], they have not been sufficiently harnessed, stimulated, or promoted by health institution [10].

The objective of this study is to analyze the implementation of the protective measures against COVID-19 in the Spanish population and to determine the factors involved in their compliance.

## 2. Materials and Method

### 2.1. Study Design and Participant Selection 

In order to respond to the proposed objectives, a cross-sectional descriptive study was carried out in which a total of 5560 people residing in Spain participated. Specifically, the study was carried out in three Autonomous Communities representative of different types of population in this country. One of the Autonomous Communities chosen was Cantabria, a mainly rural population located in the north of Spain. Another Community was Madrid, the capital of the nation, which is located in the center of the country and whose population is mainly urban. Finally, the Canary Islands, located in the south of Spain, were chosen to represent the island population.

To select the study population, two inclusion criteria were established: being at least 18 years of age and residing in Spain at the time of recruitment. Individuals who did not tick the box to accept informed consent and/or did not fully complete the study questionnaire, were excluded from the study. On this basis, of the 5602 people who participated, a total of 5560 were selected.

Snowball sampling (non-probability consecutive sampling technique) was used. Given the complexity of obtaining data during a pandemic, this method was determined to be the most effective to obtain the largest possible number of responses through the cumulative effect it generates. Figure 1 details the sampling process.

Taking into account the large sample size obtained, the results can be considered representative for each population subgroup [13,14]. 

The sample was weighted by age and sex, assigning the weights observed in Table 1. 

### 2.2. Data Collection

Data collection was conducted between 4 and 9 August 2020, after the period of mandatory confinement established in Spain.

For this purpose, an ad hoc online questionnaire was created and shared through social networks, scientific societies, and various health institutions. 

Specifically, it was distributed through virtual means of the partners and the research team (email and social networks such as WhatsApp, Instagram, Facebook, and Twitter). At the same time, collaboration was requested from the public through communications and publications on the web pages of the research partners. Finally, notifications were sent through the APP “Cantabria Salud”, requesting the population to fill in the questionnaire.

This questionnaire included the description of the project, informed consent, and four groups of questions focused on: (a) sociodemographic characteristics of the participants, (b) contact they had had with COVID-19, (c) fear and perception of risk SARS-CoV-2 infection, and (d) compliance with protective measures. 

### 2.3. Variables

The dependent variable, defined as compliance with protective measures against coronavirus, was based on the construction of an ordinal variable resulting from the sum of compliance with the recommendations established by the competent authorities. These recommendations were: (a) maintaining physical distance between people (Yes/No); (b) using a mask outdoors (Yes/No); (c) using a mask when meeting with family and friends (Yes/No); (d) using a mask when shopping (Yes/No); (e) handwashing (Yes/No); (f) using hydroalcoholic gel (Yes /No). The following dichotomous variable was created, 0 = Low/No compliance; 1 = High/Total compliance. The code 0 = “Low”, was assigned to those who complied with two recommendations or less; the code 1 = “High”, was assigned to those who complied with at least three recommendations.

As independent variables were included: gender, age, and risk of infection measured with a variable of perception that included the categories: 0 = Low; 1 = Moderate; 2 = High. The variable "threat” was defined as consideration of COVID-19 as a dangerous public health issue. This variable was asked as: “Currently, do you consider COVID-19 a dangerous public health issue?” The response categories were 0 = No; 1 = Less than before; 2 = Same as before; 3 = More than before. “Before” refers to the period between March and June 2020, the first wave of the outbreak. 

### 2.4. Analysis 

The applied variables to predict the perception of compliance with the COVID-19 prevention regulations, were defined. Subsequently, the variables were tested for independence using Pearson’s chi-square test between the independent variables and the dependent variable (compliance). The binary logistic regression analysis was performed to predict compliance with the protective measures based on the independent variables described. The analysis generated a new variable with the risk probabilities for each individual. This new variable was used in the comparison of means of the independent groups and ANOVA, to identify differences in the probability of compliance in some subsamples. The analysis of significance of the *t*-tests and the *F* tests was completed with the calculation of the effect size through Cohen’s *d* [15,16]. The *t*-test and *F* tests for ANOVA were included as complementary analyses using the probability calculated in the logistic regression as the dependent variable. These analyses allow us to see the differences between dichotomous or polytomous categories in different variables beyond the variables that were included in the logistic function.

We used the Enter method in the regression after verifying that the Forward Conditional and Backward Conditional methods provided similar results related to predictive power and goodness of fit.

In our analyses, we have used the Cohen’s *d* as a measure of effect size whose equivalencies can be observed in other magnitudes, such as r or η^2^, as proposed by Cohen [17].

### 2.5. Ethical Aspects

Favorable report from the Cantabria Clinical Research Ethics Committee, according to act 16/2020, Code 2020.159; as well as the Research Ethics Committee of the Universidad Europea de Madrid (CIPI/20/150).

The questionnaire used in the survey included information about the study (description and purpose of the study) as well as the informed consent which participants had to accept in order to participate in this study.

## 3. Results

The descriptive analysis of the variables and their categories is detailed in Table 2.

Table 3 shows the intervening variables in the regression equation and suggests a statistically significant association with the variable “Compliance with safety regulations against coronavirus”.

Other variables, such as education level, income, employment status, degree of closeness to the disease (if the participant or a family member has been affected by the disease), and living with a healthcare worker, did not show significant associations with the compliance with protective measures against the disease. 

The results of the model show a high predictive value: 81.7% (cut-off value = 0.050). The Omnibus Test of Model Coefficients shows the adequacy of the model with a *p* < 0.001 and a Chi-square value = 449.89. The goodness of fit with the Hosmer–Lemeshow test gives a value of 16.07 and a significance *p* = 0.041, indicating the fit of the model variables.

The regression coefficients of the model show statistical significance (*p* < 0.001 or *p* < 0.005), with a confidence level of 95%.

The probability of high or total compliance with the protective measures against coronavirus is higher in women (OR = 1.401) and as age increases, with an OR = 2.524 in the interval between 31 and 64 years old and an OR = 2.896 in the oldest interval (65 and over). This suggests that as age increases, there is a greater commitment to comply with protective measures. 

Risk perception also affects compliance, suggesting that those who perceive greater risk show higher compliance with the protective measures. The OR corresponding to high risk has a value of 1.423, and the OR of moderate risk has a value of 1.181. However, the most noticeable variable due to the odds ratio values is the perception of threat that COVID-19 poses as a public health issue. Compared with the group that believes there is a threat, the OR value increases in the group that considers the threat to be less of an issue than before (OR = 5.129), the same as before (14.622), and greater than before (20.952). Therefore, a person who considers that the disease poses a greater threat to public health than it did before is 20 times more likely to comply with the protective measures than a person who does not consider the disease a threat to public health (Table 4).

Following this, we conducted a comparative analysis of the means for independent samples, comparing different sample subgroups through the *t* statistic and *F* in the case of ANOVA. The results in Table 5 show that women are more likely to comply with the recommendations to stop the spread of the pandemic (0.840) compared to men (0.772) (the size effect measured with Cohen’s *d* = 0.586). Older people show a higher probability of complying (0.834), followed by those between 31 and 64 years old (0.8302). Participants under 30 years of age show a lower probability of compliance (0.679). The effect size in the ANOVA analysis for the age groups shows a Cohen’s *d* = 1.289. 

Participants who had been diagnosed with COVID-19 show a lower probability of complying with the measures (0.775) than those who have not been diagnosed (0.809), with a rather low effect size (Cohen’s *d* = 0.282). Similar data are found among those who report having a chronic illness, with a higher probability of compliance (0.825) compared to those who do not suffer any chronic illness (0.791), with an effect size of Cohen’s *d* = 0.283. Participants who have changed hygiene habits at home are more likely to comply with the new regulations (0.818) than those who have not adopted new hygiene habits (0.789), with Cohen’s *d* = 0.233. Those participants who have restricted their leisure time outside home (0.846) also show a higher probability of compliance when compared to those who have continued with their regular outdoor leisure activities, and especially with those who have increased their activities (0.711). Cohen’s effect size *d* = 1.117.

## 4. Discussion

This study shows the characteristics of the population that considers it more likely to be infected by SARS-CoV-2, thus adopting greater adherence to prevention measures. These preventive measures have proven to be a barrier to infection, especially when there have happened peaks of maximum expansion of the virus, helping to reduce morbidity and mortality [18]. Therefore, knowing which factors are associated with greater or lesser compliance with protective measures is essential to establishing effective measures against a pandemic.

### 4.1. Sociodemographic Factors and Compliance with Protective Measures

The compliance with protection standards increases with age, and older adults are more likely to comply with these regulations. Some authors suggest that older adults are one of the groups with a higher perception of threat [19,20], this fact could explain their greater compliance with protective measures. The high percentage of deaths in this group [21,22] may lead to this sense of vulnerability. On the other hand, adults under 30 years of age shoe a lower probability of compliance. These findings are being reflected in the lower number of people currently vaccinated in this age group [23,24]. 

On the other hand, women are more compliant with the regulations than men. Consistent with other studies, this finding shows that women have a higher perceived risk of infection compared to men [25,26]. This fact is contradictory to the gender consequences, which show a higher risk of severity [27] and mortality in men [28].

Although individuals with a low-income level and fewer qualifications are at a higher risk of developing acute infections [29], we found no significant association between socioeconomic situation, level of studies, income or employment status of the surveyed population and compliance with the protective measures. The relationship with poverty has not been studied in-depth in this pandemic, and only a few studies have included these variables as they are not considered clinically relevant [30], in which the situation of helplessness exposes this population group to a higher risk of infection and of developing serious forms of the disease. 

### 4.2. Perception of Risk of Infection and Compliance with Protective Measures

An association has been established between the perceived personal and family susceptibility to disease and the implementation of protective behaviors (handwashing, home hygiene, use of masks, etc.) and avoidance strategies (avoiding public places, restaurants, shops, etc.) [31,32]. This association shows the relevance of analyzing risk perception before introducing protection behaviors in the event of a pandemic [33,34,35].

### 4.3. Clinical History and Compliance with Protective Measures

Elderly and chronically ill patients were more likely to experience emotional disturbances related to protective measures, such as confinement and social isolation [36,37]. However, it is in this social profile where we find a greater monitoring of protection measures. Older patients and those with chronic diseases have the highest adherence to health recommendations to prevent infection [38]. Some authors point out that these measures generate stress and anxiety in this profile of people [36,37]. Although patients who overcome SARS-CoV-2 infection may maintain impairments, such as respiratory or cognitive problems [39], people who have already passed COVID-19 show a lower likelihood of meeting compliance measures. One of the possible reasons may be the false sense of protection after infection. However, the possibility of re-infection is a reality and there is even a likelihood of hospital readmission [40].

### 4.4. Institutional Approach to Population Behaviour

The institutional approach to population behavior has focused on risks without taking into account the capacities of communities [10]. Measures have been implemented and decisions have been taken in a unidirectional way from health institutions without listening to the population, but attempting to motivate them through fear instead of seeking conviction and commitment [10].

There are examples that show how the overwhelming pandemic situation generated community actions, such as the creation of mutual support networks in which professionals, local administrations, and municipalities took the lead in self-organizing and ensuring care [41].

### 4.5. Limitations

The main limitation of this study is that the sample was not random, which restricts the general applicability of the results. Snowball sampling was considered the most appropriate due to the exceptional circumstances of the pandemic, which precluded the use of more precise sampling methods. However, it is worth highlighting the high level of responses which, although it does not allow us to extrapolate the results to the population as a whole, does allow us to ascertain compliance with the recommendations and the profile of the population closest to these measures.

The use of a virtual medium for the distribution and collection of data is a limitation for that part of the public that does not have access to these tools. Even so, it is possible to assume a certain level of quantitative representativeness, with partially controlled biases, due to the large sample size.

In order to correct for sample bias, post-stratification was used to adjust the rough estimates. This argument is based on the results of studies, such as those of Wang, Rothschild, Goel, and Gelman, which show that, by making appropriate statistical adjustments to election polls, surveys with non-random samples can be used to generate accurate results [42]. The same authors conclude that non-representative polls are useful not only for predicting the winner of an election, but also for measuring public opinion on a wide range of social, economic, and cultural issues.

Finally, we would like to highlight that, after our study, the Italian National Epidemiological Survey on COVID-19 (EPICOVID19) was developed and validated. This specific, cross-sectional, self-administered scale was used in other studies as a tool to determine fear of COVID-19, among others [43,44], and its items closely correspond to those constructed in our design.

## 5. Conclusions

This study analyses the degree of compliance with preventive measures against COVID-19 in the period immediately after the lifting of strict containment measures in the Spanish population, i.e., after June 2020, which was referred to as the “new normal”.

The data show an association between socio-demographic variables and compliance with protective measures. Higher compliance is observed among women and those over 65 years of age. This may be related to the fact that the population over 65 years of age had the highest mortality and severity rates. The logistic model also shows that a higher perceived risk of infection and having a chronic disease favor compliance with protective measures. However, compliance with the measures is lower in people who have previously had COVID-19.

Analyzing the population’s compliance with protective measures, while encouraging their participation and collaboration in a pandemic situation, is essential for effective protective measures. In this respect, the main practical implication of the study is that it would allow the establishment of interventions and guidelines aimed at the general population and, above all, at specific population groups, which would help them to achieve a balanced understanding between real and perceived risk. This would ensure compliance with preventive measures in future epidemic situations.

Governmental awareness campaigns on COVID-19 and related protective measures should be tailored to specific segments of the population. The health administration must promote the leadership of the different communities of individuals that make up a population. This will favor the empowerment of people in terms of controlling their own health, while at the same time allowing responses to be adapted to each territory and each reality, thus guaranteeing a greater degree of compliance with preventive measures. To sum up, we believe that our findings may be useful for designing future awareness-raising campaigns adapted to different socio-demographic characteristics in COVID-19-affected settings. In addition, the present study contributes to further research on population compliance with preventive measures during a pandemic.

## Figures and Tables

**Figure 1 ijerph-19-15983-f001:**
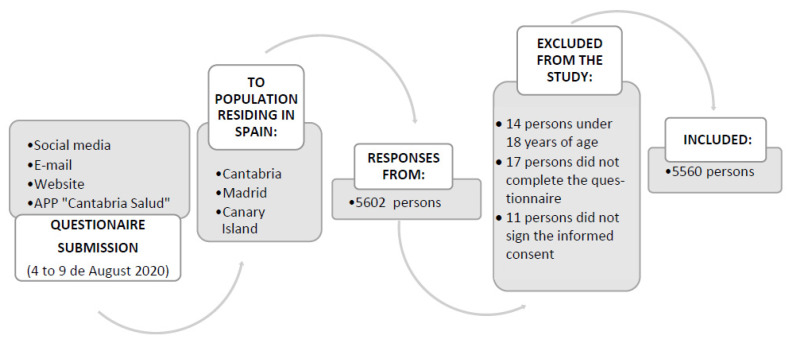
Sampling process.

**Table 1 ijerph-19-15983-t001:** Allocation of weights in the weighing of samples. 4 and 9 August 2020; Spain. Weights allocated to each substratum.

	Males	Females	Total
Up to 30 years	2.810	0.894	1.370
From 31 to 64 years old	1.238	0.532	0.742
More than 64 years old	2.610	3.892	3.207
Total	1.563	0.747	1.000

**Table 2 ijerph-19-15983-t002:** Descriptive analysis.

	*N*: 5560 % (N)
Gender	
Male	48.5 (2696)
Female	51.5 (2864)
Age	
Up to 30 years old	16.4 (910)
From 31 to 64 years old	59.9 (3328)
65 years and over	23.8 (1321)
Threat of infection (COVID-19 as a public health issue)	
It is not an issue	1.5 (85)
It is less of an issue than before	14.6 (813)
The issue is the same as before	69.8 (3883)
It is a bigger issue than before	14.0 (779)
Risk of infection (risk perception)	
Low	31.6 (1755)
Moderate	52.7 (2929)
High	15.8 (876)

**Table 3 ijerph-19-15983-t003:** Compliance with protection measures distributed according to the different variables. Tests have been performed with a CL95%. The reference category (ref.) is indicated for subsequent logistic regression analysis.

	Compliance with Protective Measures		Pearson Chi-Squared Test	Asymp. Sig. (2-Sided)	*d* Cohen
	Low/Nil	High/Total			
Age					
Up to 30 years (ref.)	32.1% (292)	67.9% (618)	113.17	<0.001	0.288
31–64 years old	17.0% (565)	83.0% (2763)			
65 years and over	16.6% (219)	83.4% (1103)			
Gender					
Male (ref.)	22.8% (616)	77.2% (2080)	41.36	<0.001	0.173
Female	16.0% (459)	84.0% (2404)			
Risk of infection					
Low (ref.)	23.9 (420)	76.1 (1335)	38.86	<0.001	0.168
Moderate	17.9 (525)	82.1 (2404)			
High	14.8 (130)	85.2 (746)			
Threat to Public Health					
It is not (ref.)	75.3% (64)	24.7% (21)	357.89	<0.001	0.525
Less than before	35.5% (288)	64.5% (524)			
Same as before	16.3% (631)	83.7% (3252)			
More than before	11.8% (92)	88.2% (686)			

**Table 4 ijerph-19-15983-t004:** Variables in the regression analysis equation.

	B	S. E	Wald	df	Sig.	Exp (B)	95% C.I. for EXP (B)	
							Lower	Upper
Female	0.337	0.072	21.92	1	0.000	1.401	1.217	1.614
Moderate risk	0.166	0.08	4.34	1	0.037	1.181	1.01	1.38
High risk	0.353	0.119	8.75	1	0.003	1.423	1.126	1.798
Minor threat	1.635	0.265	37.95	1	0.000	5.129	3.049	8.629
Same threat	2.683	0.26	106.59	1	0.000	14.622	8.787	24.331
Greater threat	3.042	0.28	117.82	1	0.000	20.952	12.096	36.291
Between 31 and 64 years	0.926	0.089	109.06	1	0.000	2.524	2.122	3.003
65 years and over	1.063	0.109	94.38	1	0.000	2.896	2.337	3.589
Constant	−2.114	0.268	62.37	1	0.000	0.121		

**Table 5 ijerph-19-15983-t005:** Comparison of means of the probability of fear of infection of specific subgroups (ANOVA).

	N	Mean	Std. Dev	Std. Error	T/F	Sig. (2-Tailed)
Gender						
Male	2696	0.772	0.134	0.003		
Female	2864	0.840	0.096	0.002	−21.666	<0.001
COVID-19 Diagnosis						
No	5214	0.8087	0.1164	0.0016	3.641	<0.001
Yes	345	0.7747	0.1709	0.0092		
Chronic illness						
No	2977	0.7909	0.1341	0.0025	−10.731	<0.001
Yes	2582	0.8247	0.1003	0.0020		
People follow the regulations outdoors						
No	4278	0.8155	0.1074	0.0016	8.385	<0.001
Yes	1281	0.7769	0.1539	0.0043		
Has changed hygiene habits at home						
No	2176	0.7886	0.1381	0.0030	−8.464	<0.001
Yes	3384	0.8181	0.1067	0.0018		
Has carried out leisure activities outside the home						
No	1347	0.8460	0.0876	0.0024	109.66	<0.001
Yes, less often than before	3789	0.8008	0.1182	0.0019		
Yes, as often as before	394	0.7347	0.1782	0.0090		
Yes, more often than before	30	0.7108	0.1698	0.0308		
Age						
Up to 30 years old	910	0.6795	0.1279	0.0042	769.27	<0.001
From 31 to 64 years old	3328	0.8302	0.1035	0.0018		
65 and over	1321	0.8346	0.0992	0.0027		

Note: Student’s *t*-test for comparing means of two independent groups; *F*-test for ANOVA when there are more than two groups. All tests have been performed for CL95%.

## Data Availability

Data available on request due to restrictions e.g., privacy or ethical.

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
