# Peer review of "Compliance with COVID-19 Prevention Measures in the Spanish Population during the New Normal: Will the Need for Greater Community Involvement Be One of the Lessons Learned?"

_ijerph, 2022, doi:10.3390/ijerph192315983_

Round 1
Reviewer 1 Report
This article, reported by Ana et al, is an interesting socio-epidemiological study that analyses compliance with preventive actions against COVID-19 in a Spanish population in the framework of a cross-sectional study and attempts to identify the factors that mediate this. While this study may be useful, there are some crucial questions and concerns that I would like to raise. Although the overall wording and paper-writing techniques are quite poor and inherently likely to result in rejection for publication, I would be happy to consider accepting the manuscript if the author is willing to answer my questions constructively and with substantial revisions.
L21-33, In this abstract, there are the words Background, Methods, Sample, Results and Conclusions - is this a accept for this journal? In general, I think these should be removed and the abstract should be written in a textual way that is easy for the reader to understand. A further point I would like to comment on is that it is observed that the text of the abstract may not be academically appropriate. For example, it is suddenly clear that “A cross-sectional study was carried out” or “An online survey was used to collect data for 6 consecutive days after the mandatory confinement period established during the state of alarm in Spain” is severely lacking in information and the flow of the sentence is not clear to the reader. Next, Results should be more specific and include numerical information on the results and the statistics of the model. In Conclusions, the beginning ”The data” shows is not satisfactory and should be corrected.
L37-39, The meaning of the sentence is very difficult to convey and should be corrected. Also, the term should not be used without defining the official name of SARS-CoV-2.
L41, Define Covid-19.
L44, The author has started to use the term COVID-19, even though Covid-19 has been used previously. What does this mean - is there a difference between Covid-19 and COVID-19? The reader doesn't know and I didn't understand it either.
L44, SARS-CoV-2 coronavirus is not normally used; shouldn't coronavirus be deleted?
L52-54, When are the first months of the pandemic - 2020, 2021 or 2022?
L55-60、I am sorry, but I do not know what you want to present. I also do not think that the purpose of this research is even presented. The abstract mentions 2020 in its entirety, the background is not stated in a defensible and comprehensive manner, and the objective is not even presented. Overall, I would like to see all the text revised to facilitate a better understanding for the reader. In particular, it is now 2022 and more than two and a half years since the outbreak of the pandemic. In light of this context, the text structure should be conscious of the time frame in which this study was conducted and the value it can present.
L61, There is a subsection in Introduction and it is not clear what is meant by discussing the following. Below this, the objectives appear to be present and discussed, but there are so many ways to divide the sections that it would be very difficult for the reader to understand.
L100-105, Please prepare and present a map describing the location of the target regions of Spain - Madrid, Cantabria and the Canary Islands. International readers are not familiar with the information on Spain, are they?
L106-113, Present the sampling method as a flowchart.
L114, What is meant by weighting? And why is Table 1 necessary? The numbers in the table don't make sense either. Moreover, they are divided into groups of up to 30, 31-64, and greater than 64. There is no explanation for this either.
L152-172, Can the authors justify why they used logistic kai ki analysis in their statistical analysis? Also, what statistic did you use to infer the association? Is it the odds or the oddity coefficient? Also, is the significance level two-sided or one-sided of the probability distribution?
Results, The description of the results is unavoidable, but the table should be changed again to something easier to understand. Few other papers or similar papers have been checked that use this type of wording, and it is too difficult to understand in this case.
Discussions and Conclusions, I think it is generally well lacking.
Reference Style, Does your bibliography follow the MDPI style, I think MDPI does not cite in the body of the manuscript in the style of, for example, (Pavia & 112 Aybar, 2018; C. Wang et al., 2020). I believe they usually use numbers such as [1]. Check here as well.
Author Response
Thank you for agreeing to review this manuscript, the contributions you have made enable this work to be improved. We respond to your contribution in the attached document.
Please see the attachment.
Comments to the editor:
The document has been revised and modifications have been included in the material and method, discussion and conclusions sections, in order to improve the different sections and eliminate similarities with previous articles.

Reviewer 2 Report
Dear Authors,
This manuscript presents a very interesting topic. However, some information is needed to improve this manuscript. Minor revisions are needed to improve it. Please see the attachment below. Thank you.

Author Response

(The authors gave the same response as above.)

Round 2
Reviewer 1 Report
Let me thank you for your effort in addressing all my comments. I am fully satisfied with your revision.